# Novel TiO_2_ Nanoparticles/Polysulfone Composite Hollow Microspheres for Photocatalytic Degradation

**DOI:** 10.3390/polym13030336

**Published:** 2021-01-21

**Authors:** Shangying Zhang, Qi Wang, Fengna Dai, Yangyang Gu, Guangtao Qian, Chunhai Chen, Youhai Yu

**Affiliations:** Center for Advanced Low-Dimension Materials, State Key Laboratory for Modification of Chemical Fibers and Polymer Materials, College of Materials Science and Engineering, Donghua University, Shanghai 201620, China; 1185099@mail.dhu.edu.cn (S.Z.); 1185078@mail.dhu.edu.cn (Q.W.); 2180397@mail.dhu.edu.cn (F.D.); 1209718@mail.dhu.edu.cn (Y.G.); qgt@dhu.edu.cn (G.Q.); cch@dhu.edu.cn (C.C.)

**Keywords:** hollow composite microspheres, polysulfone, titanium oxide nanoparticles, photocatalytic degradation materials

## Abstract

Nanosized titanium oxide (TiO_2_) material is a promising photocatalyst for the degradation of organic pollutants, whereas the difficulty of its recycling hinders its practical application. Herein, we reported the preparation of a novel titanium oxide/polysulfone (TiNPs/PSF) composite hollow microspheres by the combination of Pickering emulsification and the solvent evaporation technique and their application for the photodegradation of methyl blue (MB). P25 TiO_2_ nanoparticles dispersed on the surface of PSF microspheres. The porosity, density and photoactivity of the TiNPs/PSF composite microsphere are influenced by the TiO_2_ loading amount. The composite microsphere showed good methyl blue (MB) removal ability. Compared with TiO_2_ P25, and PSF, a much higher MB adsorption speed was observed for TiNPs/PSF microspheres benefited from their porous structure and the electrostatic attractions between the MB+ and the negatively charged PSF materials, and showed good degradation efficiency. For TiNPs/PSF composite microsphere with density close to 1, a 100% MB removal (10 mg L^−1^) within 120 min at a catalyst loading of 2.5 g L^−1^ can be obtained under both stirring and static condition, due to well dispersing of TiO_2_ particles on the microsphere surface and its stable suspending in water. For the non-suspended TiNPs/PSF composite microsphere with density bigger than 1, the 100% MB removal can be only obtained under stirring condition. The removal efficiency of MB for the composite microspheres retained 96.5%, even after 20 cycles. Moreover, this composite microsphere also showed high MB removal ability at acidic condition. The high catalysis efficiency, excellent reusability and good stability make this kind of TiNPs/PSF composite microsphere a promising photocatalyst for the water organic pollution treatment.

## 1. Introduction

With the rapid development of urbanization and industry, the problems of water pollution have been concerned. Dyes, containing unsaturated chromophore and auxochrome in their structure, are widely used in many fields, such as textile, leather, papermaking, printing, food, etc. [1,2]. However, due to their refractory structure and high toxicity to ecological environment, much more attention has been focused on the treatment of dyes-contaminated water [3]. Many technologies, including adsorption, membrane filtration, chemical coagulation/flocculation, and ionic exchange have been proposed to remove pollution from wastewater [4,5]. Nonetheless, they generally have some drawbacks, such as high energy consumption, fouling problems, and the generation of secondary contaminants as solid waste. Photocatalyst can degrade organic compound into harmless constituents, having been considered as a promising and environmentally friendly material for organic pollutant treatment [6].

Various photocatalytic material, especially semiconductor-based catalysts, such as TiO_2_, ZnO, NiO, CdS, CdSe, SiC, CuO were found to be efficient to completely eliminate pollution in polluted water [7,8,9,10,11]. Among these, TiO_2_ has been wildly used in photoelectric conversion, clean energy production, and catalytic degradation, due to its interesting electrochemical properties, chemical stability, non-toxicity, powerful anti-oxidizing power, and high benefit-to-cost ratios [12,13]. TiO_2_ P25, a commercial nanomaterial containing anatase phase and rutile phase with molar ratio of about 79:21, has been used as a photocatalyst for the decomposition of water waster. However, its small size makes it hard to be recycled, leading to a high cost and the secondary pollution [14]. Therefore, the development of recyclable TiO_2_ based photocatalyst with high activity has received much attention.

Recently, many recyclable TiO_2_- based photocatalysts have been reported. Polymers with the property of low coast, chemical resistance and ease processing, have been used as substrate to immobilize TiO_2_ to solve the problems mentioned above [15]. Chen et al. prepared TiO_2_-coated magnetic poly(methyl methacrylate) microspheres for photocatalytic degradation of p-phenylenediamine with good repeatability of photocatalytic performance [16]. Bai et al. prepared the poly-p-phenylene/TiO_2_ composite microsphere via sol–gel method, which exhibited good degradation efficiency of malachite green (98.2%), and retained their activity about 85.6% after five uses [17]. Baig et al. prepared Polypyrrole-TiO_2_ composite for photocatalytic degradation of methyl orange completely within 60 min [18]. Neghi et al. prepared polyvinyl alcohol/chitosan-TiO_2_ composite applied for photocatalytic removal of metronidazole under UV-light [19]. Xu et al. prepared the Phenolic Polymer@TiO_2_ composite to degrade rhodamine B [20]. Mahmoud et al. compared the use of suspended and attached catalysts for degradation of 2, 4-dichlorophenol (2, 4-DCP), and found that suspended catalyst showed faster degradation rate than attached catalyst due to better interaction between catalyst particles and pollutant molecules [21]. Magallanes et al. prepared floating photocatalysts based on composites of low-density polyethylene (LDPE) and TiO_2_ P25, which showed show higher activity for degradation of methylene blue than pure TiO_2_ P25 under special constraining conditions, i.e., no stirring and no oxygenation. They believed that the higher activity for floating catalyst is related the more efficient illumination and more efficient oxygenation on the water surface [22]. Although many kinds of polymer have been used as substrate for the photocatalysts, most of them suffered from the low stability under long-term UV-light irradiation [23]. Polysulfone (PSF) was stable under exposure of UV-light irradiation and cannot be degraded by ·OH radicals produced in the photocatalytic process [24]. Moreover, PSF is a hydrophobic and oleophilic material, which can preparatory concentrate organics from wastewater, making it as suitable substrates for photocatalytic materials [25,26,27]. So far, reports on PSF/TiO_2_ composite photocatalysts are limited to membrane materials [24].

In this contribution, we report the preparation and use of a kind of TiO_2_ nanoparticle (TiNPs)/PSF substrate hollow microsphere composite (donated as TiNPs/PSF) as mobile photocatalysts for wastewater treatment for the first time. The hollow structure design can improve the suspension of photocatalysts in water to make a better contact with pollutant molecules. The TiNPs/PSF were prepared by the combination of Pickering emulsification and the solvent evaporation technique [28]. The commercial TiNPs served as a Pickering emulsification for PSF and anchoring on PSF surfaces. The influence of TiO_2_ loading amount on the morphologies, properties and photoactivities of the TiNPs/PSF composite microspheres on methyl blue (MB) were studied in detail.

## 2. Experimental Section

### 2.1. Materials

Bisphenol A- Polysulfone (PSF Mw ca. 80k) was obtained from Shandong Horan Special Plastic Co., Ltd. (Weihai, China) Dichloromethane (DCM, ≥99.5%), polyvinylpyrrolidone (PVP, K30) and polyvinyl alcohol (PVA) were purchased from Sinopharm Chemical Reagent Co., Ltd. (Huangpu, Shanghai, China) 1-methyl-2-pyrrolidone (NMP, ≥99.5%) was applied by Shanghai Lingfeng Chemical (Jinshan, Shanghai, China). Methylene blue (≥70%) was obtained from Aladdin Chemical (Fengxian, Shanghai, China). Phenol (Standard for GC, ≥99.5%) was provided by Maclean Chemical ((Pudong, Shanghai, China)) Titanium dioxide (TiO_2_, P25) was acquired from Degussa AG Co., Ltd. (Frankfurt, Germany). All reagents were used without further purification.

### 2.2. Preparation of TiNPs/PSF Composite Microspheres

TiNPs/PSF microspheres was prepared by the combination of Pickering emulsification and the solvent evaporation technique. Typically, 0.125 g PSF, 0.1 g polyvinylpyrrolidone (PVP) were dissolve in 6 mL NMP/DCM (1/5, volume ratio) mixed solvent, then various amount of P25 TiO_2_ (0.01 to 0.11 g) were dispersed in solvent. The mixture was stirred for 30 min, then slowly injected into 50 mL coagulation bath (1% PVA aqueous solution) through a microchannel, and kept being stirred at room temperature for 5 h. The TiNPs/PSF composite microspheres obtained were collected by filtration through a 500-mesh filter cloth. Then, washed and filtered with distilled water and ethanol to remove unattached P25 particles. Eventually, the product was dried at 60 °C for 12 h. The obtained products were named as TiNPs/PSF-x, where x means the mass ratio (0.08–0.88) of TiO_2_ to PSF in the solution. The PSF microspheres were also prepared under the same condition without the adding of P25.

### 2.3. Characterizations of TiNPs/PSF Composite Microspheres

The morphology and elemental composition study of the TiNPs/PSF composite microspheres were performed on the scanning electron microscopy (SEM, JEOL-F100, Tokyo, Japan) and energy dispersive spectrometer (EDS, Oxford, UK). The cross-section morphologies of the TiNPs/PSF composite microsphere were observed by the 3D digital microscope (HIROX RH-2000, Tokyo, Japan). The cross-section morphology was prepared by imbedding the microspheres in epoxy resin (Epoxy 154+ triethylenetetramine 10:1 prepare model), which was grinded and polished by metallographic machine using 4000 mesh sandpaper (QATM Saphir, 550, Mammelzen, Germany). UV–Vis spectrophotometer (PerkinElmer Instrument, 1901 Lambda 950, Waltham, MA, USA) was employed to detect the UV–visible absorption spectra of MB during adsorption/ photocatalysis reaction and PSF, TiNPs/PSF and TiO_2_ diffuse reflection analysis. Thermal gravimetric analysis (TGA, TA Instruments, DISCOVERY TGA550, New Castle, DE, USA) of composite microspheres were recorded under a heating rate of 10 °C min^−1^ in air atmosphere to analysis TiO_2_ content in TiNPs/PSF composite microsphere. The crystalline phases were investigated by X-ray diffractometer (XRD, RIGAKU, D/max2550VB, Tokyo, Japan). The XRD patterns were recorded in the 2 θ range of 10–90° with a scanning speed of 2° min^−1^. The TiNPs/PSF composite microspheres, PSF microspheres and TiO_2_ nanoparticles were characterized by FTIR spectroscopy using KBr pellet pressing method (Bruker, Vector 22, Karlsruhe, Germany). The valence state and functional groups or typical chemical bonds of the composite microsphere was analyzed by X-ray photoelectron spectroscopy (XPS, Thermo Scientific, Escalab 250Xi, Waltham, MA, USA). The porosity of the TiNPs/PSF composite microsphere was assessed by the classic gravimetric method and the density was calculated by the Equation (2). Each sample was divided into three tested, and the result was the average of three replicates.

The porosity measurement is based on gravimetric method, which is calculated as following Equation (1):(1)$$P\left(\%\right)=\frac{\frac{W_{A}-W_{B}}{\rho_{W}}}{W_{B}/\rho_{P}+\frac{W_{B}-W_{A}}{\rho_{W}}}\;\times \;100$$

The density calculation formula is as following Equation (2):(2)D=1−PWTρT+WPρP
where *W_B_* and *W_A_* are the weight of the microsphere before and after drying, respectively. *W_T_* and *W_P_* are the weight of TiO_2_ and PSF in composite microsphere, respectively. *ρ_W_* (1.0 g cm^−3^), *ρ_P_* (1.24 g cm^−3^) and *ρ_T_* (4.23 g cm^−3^) is the density of water, polysulfone, and TiO_2_, respectively.

### 2.4. Adsorption Performance of TiNPs/PSF Microsphere

The adsorption capacity of TiNPs/PSF microspheres for methylene blue (MB) were evaluate 50 mg of TiNPs/PSF-0.40 was added into 20 mL MB solution (10 mg L^−1^) under vigorous stirring 30 min in dark at 25 °C. Then the solution was collected and tested under UV-Vis spectrophotometer every 10 min. The instantaneous adsorption capacity (*q*_*t*_) of the TiNPs/PSF were determined by the change of MB concentration, and calculated by the following Equation (3):(3)$$q_{t}=\left(\frac{C_{r0}-C_{t}}{m}\right)\times V$$
where *C_r_*_0_ (g L^−1^) and *C_t_* (g L^−1^) are the initial MB concentration and concentration at time *t*, *V*, (L), and *m* (g) represent the volume of dye solution and the weight of adsorbent, respectively.

For comparison, the adsorption experiments were also conducted by using 50 mg TiO_2_ and 50 mg PSF microspheres.

### 2.5. Photocatalytic Activity Measurement of TiNPs/PSF Composite Microspheres

Before the reaction, 50 mg catalyst was kept in the MB solution (20 mL, 10 mg L^−1^) in the dark with vigorous stirring for 30 min to reach adsorption equilibrium. Then, the suspension was exposed under ultra-violet (250 W mercury lamp) irradiation for photocatalytic degradation. A total of 1 mL reaction solution was collected at regular intervals of 30 min and centrifuged to obtain a clear solution. The degradation efficiency was monitored by UV-vis spectrophotometer. The percentage of the MB degradation rate was calculated from the following Equation (4):(4)$$Degradation\left(\%\right)=\left(C-C_{r0}\right)/C_{r0\;}\times 100$$
where *Cr*_0_ and *C* are the concentrations of the dye at irradiation times t = *0* and t = *t*, respectively.

After reaction, the TiNPs/PSF composite microspheres were recovered by centrifuged, washed with deionized water, dried and reused for the next cycle of reaction test.

## 3. Results and Discussion

### 3.1. Characterization of TiNPs/PSF Microsphere

The preparation of TiNPs/PSF composite microspheres involved the process combining Pickering emulsification and the solvent evaporation, as illustrated in Scheme 1. First, the PSF, PVP and TiNPs was dispersed in NMP/DCM (1/5, volume ratio) to form the oil phase. Then the oil phase was extruded through a microchannel into the water phase (1% PVA aqueous solution) with continuous stirring to form TiNPs-stabilized oil/water Pickering emulsion droplets [28]. Meanwhile, the exchanging of solvent with water, and the evaporation of DCM induces the phase separation of homogeneous solution inside the droplets. Since water had a higher surface tension than the polymer, it tended to migrate and coalesce inside the droplet solution rather than outside the droplet surface, occupying the internal space to form a continuous liquid core inside the microsphere during the phase separation [29]. With the evaporation of DCM, PSF precipitated to form a hollow microsphere structure eventually.

The morphology of PSF microspheres and TiNPs/PSF composite microspheres were characterized by SEM, as illustrated in Figure 1a–g. Both small and larger PSF microspheres with smooth surface can be observed, and the diameters of the microspheres are ranging from 29 to 50 μm. Some PSF microspheres is broken, showing a hollow corn structure with the porous shell about 2.2 μm. During the phase separation process, while most water merged to form the continuous core, some water stayed in the polymer phase to form a porous structure on the shell. Different from the PSF microspheres, the surfaces of TiNPs/PSF composite microspheres were rougher. With the increasing adding amount of P25 in the precursor solution, the surfaces of microspheres were gradually covered by TiNPs completely, and the shape of these microspheres became more irregular. All of the microspheres were intact, and no broken ones can be found. To explore their internal structure, ultrasonic treatment was tried to wreck the composite microspheres first, but failed. Then the composite microspheres (TiNPs/PSF-0.40) were successfully wrecked by grinding and polishing with metallographic machine. The cross-section optical micrograph of the microsphere (Figure 1h) and its 3D stereo imaging (Figure 1i) confirm that the composite microsphere also possessed hollow structure. Compared with the polymer microsphere, the composite microsphere had a much thicker shell of 15 μm, which could explain the better mechanical strength it showing. The SEM image of the inner surface of the composite microsphere showed, similar to the polymer microspheres, the shell had a porous structure. No obvious accumulation of TiNPs can be observed. The EDS analysis also confirmed that the TiO_2_ content in the inner surface is quite low (Appendix A). These SEM images clearly reveal that, TiNPs were indeed served as a stabilizer for the Pickering emulsion by adsorbing at the oil/water interface, and the hollow porous microspheres was formed by the phase separation induced by the both the evaporation of DCM and the exchanging of solvents with water.

To quantify the loading amount of TiNPs on TiNPs/PSF-x composite microspheres, the TGA analysis was conducted, and the curves were displayed in Figure 2a. For all TiNPs/PSF-x composite microspheres, the initial decomposition temperature is 350 °C, and the significant weight loss is observed in the temperature range of 450−600 °C, which is ascribed to the decomposition of PSF. The mass fraction of TiNPs in TiNPs/PSF composite microspheres are calculated as 7 wt.%, 16 wt.%, 22.5 wt.%, 28.5 wt.%, 33.5 wt.%, and 35.4 wt.% for x as 0.08, 0.24, 0.40, 0.56, 0.72, 0.88, respectively. The result showed that, the loading amount of TiNPs in the TiNPs/PSF composite microspheres increased with the adding amount TiNPs in the synthesis mixture. However, not all of the TiNPs can be incorporated onto the PSF microspheres surface by the forming stable oil/water Pickering emulsion, especially when the adding amount TiNPs is high.

The porosity and density of the TiNPs/PSF composite microspheres were analyzed and shown in Figure 2b and Table 1. With the increasing of x value from 0.08 to 0.88, the porosity of TiNPs/PSF-x decrease from 35.2% to 24.1%, and the density increases from 0.846 to 1.254 g cm^−3^, respectively. The decreasing of porosity of TiNPs/PSF composite microspheres with the increasing of TiNPs was because of the formation of less porosity structure by the increasing solution viscosity and the phase separation delay as a result of adding TiNPs [30,31]. The density change of TiNPs/PSF composite microspheres were related to the change of porosity and the amount of TiNPs. Since the porosity decreased with the additional amount of TiNPs, and the density of TiNPs is higher than PSF, it is quite easy to understand the densities of the TiNPs/PSF composite microspheres increased as TiNPs adding amount increased. The results of 2h settlement resistance test for TiNPs/PSF-x composite microspheres are shown in Figure 2e. Due to the density of the TiNPs/PSF-0.40 being closer to 1, it can suspend well in in water.

Figure 2c displays the XRD pattern of PSF, P25 TiO_2_ and TiNPs/PSF-0.40 composite microspheres. PSF shows a board diffraction peak at 2 θ of 17.89°. P25 TiO_2_ exhibits several characteristic peaks belonging to anatase TiO_2_ (JCPDS cards No. 00-021-1272) at 2 θ of 25.24° (101), 37.84° (004), 48.13° (200), 53.88° (105), 55.02 (211), and 62.82° (204), 68.76° (116), 70.30° (220), 75.03° (215), 82.66° (224), and a characteristic peaks of rutile TiO_2_ (JCPDS cards No. 00-021-1276) at 2 θ of 27.44° (110) [32,33]. The above-mentioned diffraction peaks can all be found in the XRD pattern of the TiNPs/PSF-0.40 composite, showing the successful immobilization of P25 on PSF.

The chemical structures of the TiNPs/PSF-0.40, PSF and TiO_2_ were characterized by FTIR spectroscopy. As shown in Figure 2d, the broad absorption band from 3400 to 3200 cm^−1^ and the weak peak at 1616.3 cm^−1^ in the spectrum of the P25 TiO_2_ are attributed to the stretching vibration of hydroxyl groups (OH) and absorbed water on the surface [20]. The FTIR spectra of the peaks situated at 1150 cm^−1^, 1242 cm^−1^, and 2967 cm^−1^ can be assigned to the −SO_2_ stretch, C−O−C linkage, and aliphatic C−H stretch, respectively. This implies the presence of PSF [34]. A slight shift around 3400 cm^−1^ ascribed the stretching vibration of hydroxyl groups exhibits a red shift, from 3392 to 3386 cm^−1^, which reveals the increase of hydrogen bond interaction on the TiO_2_ surface [30,35]. Based on the above-mentioned interactions, we can expect hydrogen bond between PSF matrix and TiO_2_ nanoparticles.

Figure 3a–e shows the XPS spectra of the TiNPs/PSF-0.40 and Appendix A shows the XPS spectra of the PSF and TiO_2_. The XPS survey spectrum of TiNPs/PSF-0.40 (Figure 3a) displays the Ti 1s (565.9 eV), O 1s (531.2 eV), Ti 2p (458.5 eV), C 1s (285.1 eV), S 2p (169.2 eV), Ti 3s (62.6 eV), and Ti 3p (36.4 eV) signals, which can be observed in the XPS spectra of the PSF and TiO_2._ In the high-resolution S 2p spectrum, two peaks at 167.8 and 169.1 eV (Figure 3b) can be assigned to the C=S and S=O=S, respectively. The O 1s XPS spectra of TiNPs/PSF and PSF (Figure 3c) is separated into three peaks at 529.7 eV, 531.0 eV, 531.9 eV, and 533.4 eV, which are assigned to Ti–O–Ti (lattice O), Ti–OH (surface hydroxyl), the O=S=O and C–O species of the TiNPs/PSF microsphere, respectively. [36,37] The XPS spectra of C1s (Figure 3d) of is fitted into four peaks of 284.6 eV (C–S) and 286.0 eV (C–C) [24]. The peaks situated at 458.2 eV and 464.0 eV can be assigned to the 2p_3/2_ and 2p_1/2_ of Ti^4+^, correspondingly (Figure 3e). The FTIR and XPS spectra results confirms that the TiO_2_ nanoparticles have been successfully immobilized on PSF microspheres.Figure 4a displays the UV–vis diffuse reflectance spectra pattern of P25 TiO_2_, PSF, and TiNPs/PSF-0.40 composite microspheres. The TiO_2_ showed a broad absorbance band in the UV–vis region with the maximum absorbance peak at about 340 nm. The composite microsphere retains the similar maximum absorbance peak, and the PSF microsphere have no obvious absorption peak. At the same time, the band gap energy measurements using Tauc plot for TiO_2_ and TiNPs/PSF composite microsphere. The optical absorption of a crystalline semiconductor near the energy band edge can be expressed by the following formula [18]:(5)$$Ahv=K{\left(hv-E_{g}\right)}^{n}$$
were *A* is the absorption coefficient, *K* is the proportionality constant, *h* is the Planck constant (h = 4.136 × 10^−15^ eV s), *v* is the light frequency, and *E_g_* is the band gap. TiO_2_ is an indirect band gap semiconductor, the calculated values of n for TiO_2_ and TiNPs/PSF are set at 2 [18]. The band gap energy estimated from Figure 4b,c for TiO_2_ and TiNPs/PSF composite microsphere are 3.31 eV and 3.24 eV, respectively.

### 3.2. MB Removal by TiNPs/PSF Microsphere

The dyes removing performance of TiNPs/PSF composite microspheres were evaluated by the adsorption and photodegradation experiment. For the adsorption experiment, the TiNPs/PSF microspheres, PSF microsphere, as well as TiO_2_ P25, were added into the MB solution for 30 min, to achieve an equilibrium in the adsorption-desorption of MB on them. The initial concentrations of MB were 10 mg L^−1^, and Appendix A summarized the instantaneous adsorption capacity (*q_t_*) of these samples determined by the change of MB concentration every 5 min. As shown in Figure 5a, the adsorption capacity of TiNPs/PSF composite microsphere and PSF microsphere increase rapidly at the initial stage, and then slowed down until it reached equilibrium. Compared with the TiO_2_ P25, TiNPs/PSF and PSF microspheres showed much higher adsorption speed benefited from porous structure and the electrostatic attraction between the MB^+^ and the negatively charged PSF materials [38,39]. Therefore, the quasi-first-order model (Figure 5b), quasi-second-order model (Figure 5c), and intra-particle diffusion model (Figure 5d) models were used to analyze their adsorption capacity, which is calculated as following equation [40]:(6)$$\mathrm{pseudo}-\mathrm{first}-\mathrm{order}:\;ln\left(q_{e}-q_{t}\right)=lnq_{e}-k_{1}t$$
(7)$$\mathrm{pseudo}-\mathrm{first}-\mathrm{order}:\;\frac{t}{q_{t}}=\frac{1}{k_{2}q_{e}^{2}}+\frac{t}{q_{e}}$$
(8)$$\mathrm{interparticle}\;\mathrm{diffusion}:\;q_{t}=k_{3}t^{0.5}$$
where *q_t_* (g mg^−^^1^) and *q_e_* (g mg^−^^1^) are the adsorbed amount at time t and at equilibrium, respectively. k_1_ (min^−^^1^), k_2_ (g mg^−^^1^ min^−^^1^) and k_3_ (mg g^−^^1^ min^−^^0.5^) are the rate constants of pseudo-first order, pseudo-second order and intraparticle diffusion model, respectively. t (min) is the adsorption time. The corresponding spectra is shown in Figure 5 and the relevant values are shown in Table 2. The R^2^ from the model which indicated that the adsorption data of TiNPs/PSF, TiO_2_ and PSF were more fitted with the pseudo-first-order model than pseudo-second-order model. The k_1_ calculated from pseudo-first order model indicated that the adsorption speed of PSF and TiNPs/PSF were more than pure TiO_2_, and the reason that TiNPs/PSF is slightly higher than PSF is that the rough surface of the composite microspheres provides more adsorption sites. For the intraparticle diffusion model, the regression line of TiO_2_ through the origin exhibit excellent degree of fit (R^2^ = 0.9876), which showed that the diffusion of MB molecules from the solution to the surfaces of TiO_2_. The regression line of TiNPs/PSF composite and PSF microsphere does not pass through the origin. The reason of this phenomenon may be the existence of pores on the surface of PSF microspheres and TiNPs/PSF composite microspheres, which could cause the initial boundary layer diffusion effects and the unique mass transfer in the late pores [4].

After the adsorption experiment, the MB solution were subjected to the UV irradiation with stirring for 120 min. The degradation process as shown in Scheme 2. Under ultraviolet light, the electron transition occurs on the surface of the TiNPs/PSF composite microspheres, photoelectrons (e^−^), and holes (h^+^) react with O_2_ and H_2_O respectively to generate O_2_^−^ and ·OH. The strong oxidizing radicals and holes could decompose MB into harmless constituents. With the presence of TiNPs/PSF composite microspheres, all of these blue solutions changed into colorless solution after 120 min with a degradation degree of 94.7–100.0% (Figure 6a). Except TiNPs/PSF-0.08 and TiNPs/PSF-0.24, all the TiNPs/PSF composite microspheres can completely degrade the residual MB, and the degradation results were summarized in Table 3.

The adsorption and photodegradation of MB were also studied under static condition. Two samples, suspended TiNPs/PSF-0.40 and non-suspended TiNPs/PSF-0.88, were tested and the results are shown in Figure 6b. Although TiNPs/PSF-0.88 can degrade 100.0% MB within 120 min at stirring condition, it can only degrade 64.3% MB under non-stirring state. The lower efficiency under non-stirring state is attributed to a part of catalyst particles settled at the bottom of the reactor, which decrease the utilization rate of catalyst, and the result is coherent with reported in the literature [41]. While, for TiNPs/PSF-0.40, the degradation effect can reach 100.0% with or without stirring, because TiNPs/PSF-0.40 composite microspheres can suspend well in water. Moreover, the photocatalytic activity of the TiNPs/PSF-0.40 composite microspheres can retain 96.5% degradation rate under non-stirring state after 20 cycles (Figure 6c). Compared with the prior reports of the photocatalytic degradation of MB by polymer/TiO_2_ composites, our TiNPs/PSF composite microsphere exhibits excellent catalytic effect and reusability (Table 4).

The TiNPs/PSF composite microspheres have good mechanical stability. Take TiNPs/PSF-0.40 as an example, the morphology of the composite microspheres is barely changed after the 20 cycles of degradation, as shown in Figure 7a,b. The EDS studied also proved that, only slightly loss of TiNPs on the surface of the TiNPs/PSF composite microspheres happened after degradation, showing good combination of TiNPs and PSF. The TiNPs/PSF also maintains a good morphology and most of TiNPs even after ultrasonic treatment (650 W ultrasonic 60 min, Figure 7c). In order to quantify the loading amount of TiNPs on TiNPs/PSF composite microspheres, TGA analysis was conducted, and the curves were displayed in Figure 8. The mass fraction of TiNPs in TiNPs/PSF composite microspheres are calculated as 22.5 wt.%, 18.0 wt.%, and 16.0 wt.% for as-made, after 20 cycles of degradation 0 times, and after ultrasonic treatment, respectively. The excellent mechanical properties of this TiNPs/PSF composite microspheres and the strong combination of PSF and TiO_2_ could be the reason for the good recyclability of the catalyst.

Moreover, as shown in Figure 9a, TiNPs/PSF can maintain good morphology after soaking in 3N HCl for 1 h. For TiNPs/PSF-0.40, the degradation rate of MB is 93.8% (Figure 9b) under the acid condition (PH = 1), showing good organic pollutant removal ability in harsh environment. At the acidic conditions, the negatively charged surfaces of the TiNPs repelled the negatively charged sulfonic group of the MB, leading to a decreasing of activity [42]. The above studied showed that, this new kind of TiNPs/PSF composite microsphere is a kind of promising photocatalyst for water organic pollutant treatment.

The TiNPs/PSF composite photocatalyst prepared here showed high efficiency and good recycle ability for the degradation of pollutants in water, even in the harsh condition. However, the TiNPs tended to aggregate and their utilizing efficiency is low. Further work should try to improve the dispersion of TiNPs on the microsphere surface to lower the cost for the practical application.

## 4. Conclusions

A novel kind of titanium oxide/polysulfone (TiNPs/PSF) composite hollow microsphere was prepared by a simple method combining Pickering emulsification and solvent evaporation. TiNPs served as a stabilizer for the Pickering emulsion by adsorbing at the oil/water interface, and the hollow porous microspheres was formed by the phase separation induced by the both the evaporation of solvent and the exchanging of solvents with water. The resulting hollow microspheres with thick walls of PSF and well dispersing of TiO_2_ nanoparticles the surface, showed good mechanical stability and good photodegradation ability of methyl blue (MB) under ultraviolet light irradiation. The porosity and density of the TiNPs/PSF composite microsphere can be adjusted by TiO_2_ loading amount. With the increasing of TiO_2_ adding amount value (x) from 0.08 to 0.88, the porosity of TiNPs/PSF-x decreased from 35.2% to 24.1%, and the density increases from 0.846 g to 1.254 g cm^−3^, respectively. Compared with the TiO_2_ P25, and PSF, TiNPs/PSF microspheres show much higher adsorption speed benefited from porous structure and the electrostatic attraction between the MB^+^ and the negatively charged PSF materials. Unlike the non-suspended TiNPs/PSF-0.88, which can only show high MB removal efficiency under stirring condition, the composite microsphere TiNPs/PSF-0.40 with density similar to water showed the same MB removal efficiency of 100% under stirring and static conditions. Moreover, its removal efficiency of MB retained 96.5%, even after 20 cycles, and it also showed 93.8% MB removal efficiency under acidic conditions. This new kind of TiNPs/PSF composite microsphere are expected to have applications in practical water organic pollutants treatment, due to its easy preparation, high mechanical and chemical stability, high efficiency, and good reusability.

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
