# Peer review of "Novel TiO2 Nanoparticles/Polysulfone Composite Hollow Microspheres for Photocatalytic Degradation"

_polymers, 2021, doi:10.3390/polym13030336_

Round 1

Reviewer 1 Report

This manuscript needs major revision for publication in “Polymers” Journal.

  1. In section 2.2. (on page 5), method of preparation of TiNPs/PSF composite microspheres is not clear. How uninvolved TiNPs were collected by filtration from TiNPs/PSF composite microspheres? How PSF were used or dissolved or which form of PSF were used for the TiNPs/PSF composite microspheres?
  2. The porosity measurement should be with characterization section (i.e. section 2.5), and this section (section 2.5: Characterizations of TiNPs/PSF composite microspheres) should be just before section 2.3 (Adsorption performance of TiNPs/PSF microsphere) and 2.4 (Photocatalytic activity measurement of TiNPs/PSF composite microspheres).
  3. Comparative XPS analysis of PSF, TiNPs and TiNPs/PSF composite microspheres must be included in the manuscript as the XRD and FTIR analysis were done for PSF, TiNPs and TiNPs/PSF composite microspheres.
  4. UV-DRS and bandgap analysis of TiNPs and TiNPs/PSF composite microspheres should be included in the manuscript. Because UV-DRS analysis and bandgap measurements are very important for photocatalytic studies.
  5. On page 4, there are many typographical mistakes as “duringadsorption” should be “during adsorption”, so the manuscript should be checked thoroughly for such corrections.
  6. Introduction part could be improved by adding more relevant recent literature on polymer or polymeric-inorganic based photocatalytic materials (Nanomaterials 2020, 10(6), 1098; Journal of Photochemistry and Photobiology A, 390 (2020) 112266; Colloids and Surfaces A, 611 (2021) 125886; Journal of Photochemistry and Photobiology B 204 (2020) 111783; Journal of Materials Science & Technology 33 (2017) 547-557). Aforementioned study will provide up-to-date literature review.
  7. The English needs to be improved. Get its English edited very carefully.
  8. State main findings in the conclusion.
  9. Please check the references’ format and adhere as per the style of the “Polymers & MDPI” format.

Reviewer 2 Report

This work reported the preparation and use of TiO2 nanoparticle (TiNPs)/polysulfone (PSF) substrate hollow microsphere composite as mobile photocatalysts for wastewater treatment. They were prepared by the combination of Pickering emulsification and the solvent evaporation technique. The influence of TiO2 loading on the morphologies, properties and photoactivities of the TiNPs/PSF composite microspheres was investigated. This paper can be accepted after the following revisions:

1- In authors' institute information, please double check "* Correspondence: for Advanced" and "* Correspondence: author.* Correspondence: author.".

2- The Abstract should mention more water organic pollution removal performance of the composite materials presented from this work.

3- Please compare the different wastewater purification methods in Introduction. Some related works should be considered (Mixed Dye Removal Efficiency of Electrospun Polyacrylonitrile–Graphene Oxide Composite Membranes; A review on reverse osmosis and nanofiltration membranes for water purification; etc.). What are the major advances by using photocatalysis? It is not clear that why did you use this technique for water purification.

4- Please add the comparison between TiO2 and other metal oxides in Introduction.

5- Please improve the English writing carefully throughout the work such as "chemical resistance and ease processing, has been used as" -- "chemical resistance and ease processing, have been used as" on Page 1, "Where WB and WA" -- "where WB and WA" on Page 3, "spectroscopy. As shown in Fig. 2d. The" on Page 6, “3(a-e).” -- “3(a-e)" on Page 7, "mL-1" in Table 3, etc.

6- From Refs. [8] - [12], what are the shortcomings from these cited works? Also, there is no comparison between PSF and other polymers. What are the other commonly used substrates? The authors need to discuss the status of the different substrates used during the photocatalysis process.

7- In the last paragraph of Introduction, please do not give the full term of PSF again. Please check the whole text carefully.

8- In Section 2.2, what are the improvements in the preparation of TiO2 nanoparticle/polysulfone composites? There are several publications reported on this composite (Preparation and properties of polysulfone/TiO2 composite ultrafiltration membrane; Fabrication and Evaluation of Functionalized Nano-titanium
Dioxide (F-NanoTiO2)/ polysulfone (PSf) Nanocomposite
Membranes for Gas Separation; etc.). 

9- Did you measure the densities of water, polysulfone, and TiO2?

10- In Section 2.3, how long for the stirring?

11- Section 2.5 should be moved to Section 2.3 which can be corresponded to Section 3. I suggest to move Scheme 1 in Section 2.2.

12- In Table 1, do you mean weight ratio for "mass ratio"? The unit of density is incorrect (g−1 cm−3). How could you verify the density values?

13- In Table 2, please delete the data at 0 min. Please change the unit of the adsorption capacity (remove 10−1). Please specify the adsorption speed differences from Table 2.

14- This work is mainly based on the characterizations. Any water organic pollutant treatment results for this material?

15- As the authors claimed that this is a novel kind of TiNPs/PSF composite. However, as mentioned earlier, this composite has been widely reported, even the preparation is not new in this work. The novelty and the applications of this composite should be highlighted.

16- Please check the format of the references in this wok such as "J Membrane" in Ref. [29].

In all, this paper needs major revisions before next round of peer review.

Reviewer 3 Report

The manuscript entitled Novel TiO2 Nanoparticles/ Polysulfone Composite Hollow Microspheres for Photocatalytic Degradation submitted to Polymers Journal.

The concept of the manuscript is novel, fits and suitable to publish in Polymers Journal. This manuscript is generally well written and clearly presented however still need to address many comments and thus require substantial major revision before its acceptance.

  • As this manuscript is submitted to Polymers journal so major focus should be on importance of polymers in the development of photocatalyst need to discuss in detail.
  • Provide a nice graphical abstract representing the overview of the MS with key highlights. Give full form of abbreviation in the abstract as well as in whole manuscript.
  • In the introduction section, write the novelty of the work and the problem statement clearly. Substantial discussion about the textile dye pollution and their environmental impact for this refer and cite related to dye Journal of basic microbiology 49 (S1), S36-S42, 2009. The defined research objectives should be mentioned at the end of introduction.
  • In addition to this detailed discussion of current photocatalytic studies using different approaches Journal of environmental management 223, 1086-1097, 2018; Journal of Alloys and Compounds, 155083, 2020 also need to mention somewhere in the introduction section.
  • Using UV lamp for photocatalytic studies is this useful for practical application give details of the developed economics of the process.
  • Give details of reaction mechanism of developed photocatalytic process by adding a new figure.
  • Statistical analysis of the results should be provided in the materials and methods section. It's important for all experimental work Report these values in the results and discussion.
  • What about the toxicity of the degraded products give your opinion about the same.
  • Write the practical applications and future research perspectives and challenges by adding a new section before conclusions
  • The conclusion of the study is not discussed with the specific output obtained from the study, it could be modified with precise outcomes with a take home message.
  • English and grammar mistakes are present. The author should check the manuscript by native English Speaker to improve the quality of the manuscript.

Round 2

Reviewer 1 Report

After careful analysis of the revised manuscript, I recommend for publication in Polymers Journal.

Reviewer 2 Report

It is ok to accept this work.

Reviewer 3 Report

Authors have substantially revised the manuscript according to the reviewer comments. The present form of the manuscript can be accepted.